# Functional genomic screening reveals asparagine dependence as a metabolic vulnerability in sarcoma

Simone Hettmer[1,2,3,4,5], Anna C Schinzel[6,7], Daria Tchessalova[1,5], Michaela Schneider[4], Christina L Parker[8,9], Roderick T Bronson[10], Nigel GJ Richards[11], William C Hahn[6,7], Amy J Wagers[1,5]*

[1]Department of Stem Cell and Regenerative Biology, Harvard Stem Cell Institute, Harvard University, Cambridge, United States; [2]Department of Pediatric Oncology, Dana-Farber Cancer Institute, Boston, United States; [3]Pediatric Hematology/ Oncology, Charité, University Hospital Berlin, Berlin, Germany; [4]Division of Pediatric Hematology and Oncology, Department of Pediatric and Adolescent Medicine, University Medical Center Freiburg, University of Freiburg, Freiburg, Germany; [5]Joslin Diabetes Center, Harvard Medical School, Boston, United States; [6]Department of Medical Oncology, Dana-Farber Cancer Institute, Boston, United States; [7]Broad Institute of Harvard and MIT, Cambridge, United States; [8]Summer Honors Undergraduate Program, Harvard Medical School, Boston, United States; [9]University of Maryland, Baltimore County, Baltimore, United States; [10]Department of Biomedical Sciences, Tufts University Veterinary School, North Grafton, United States; [11]Department of Chemistry and Chemical Biology, Indiana University - Purdue University Indianapolis, Indianapolis, United States

*For correspondence: amy. wagers@joslin.harvard.edu

**Abstract** Current therapies for sarcomas are often inadequate. This study sought to identify actionable gene targets by selective targeting of the molecular networks that support sarcoma cell proliferation. Silencing of asparagine synthetase (ASNS), an amidotransferase that converts aspartate into asparagine, produced the strongest inhibitory effect on sarcoma growth in a functional genomic screen of mouse sarcomas generated by oncogenic *Kras* and disruption of *Cdkn2a*. ASNS silencing in mouse and human sarcoma cell lines reduced the percentage of S phase cells and impeded new polypeptide synthesis. These effects of ASNS silencing were reversed by exogenous supplementation with asparagine. Also, asparagine depletion via the ASNS inhibitor amino sulfoximine 5 (AS5) or asparaginase inhibited mouse and human sarcoma growth in vitro, and genetic silencing of ASNS in mouse sarcoma cells combined with depletion of plasma asparagine inhibited tumor growth in vivo. Asparagine reliance of sarcoma cells may represent a metabolic vulnerability with potential anti-sarcoma therapeutic value.

## Introduction

Soft-tissue sarcomas (STS) are a heterogeneous group of non-hematopoietic, mesodermal cancers. Certain STS types present with tissue-specific features, such as skeletal muscle differentiation in rhabdomyosarcoma (RMS) (*Parham and Barr, 2013*). For most STS tumors, cure depends on radical resection and/or radiation of the tumor, and therapeutic options for tumors that have spread region-ally and/or systemically are limited (*Linch et al., 2014*). The genetic spectrum of STS is heteroge-neous. Many tumors carry complex karyotypes with variable genetic changes; others express specific

**eLife digest** Sarcoma is a type of cancer that forms in the connective tissues of the body, such as bone, cartilage, muscle and fat. Usually, treatment involves surgical removal of the tumor and/or radiation to kill the tumor cells. However, if sarcomas spread to other parts of the body, the treatment options are limited.

Genetic studies have revealed several genetic changes that contribute to the formation of sarcomas. Many sarcomas have a mutation in a gene that encodes a protein called Ras. In 2011, researchers found that injecting Ras mutant muscle cells into the muscles of mice could lead to the formation of sarcomas. Next, the researchers compared gene expression in the mouse sarcoma cells with gene expression in normal mouse muscle cells and found that certain genes appeared to be more highly expressed in the sarcoma cells. These genes were also hyperactive in human sarcoma cells and may promote the growth of sarcomas carrying mutant forms of Ras.

Now, Hettmer et al. – including some of the same researchers involved in the earlier work – show that targeting one of these hyperactive genes can slow sarcoma growth. The experiments made use of a technique called ribonucleic acid interference (or RNAi for short) to specifically switch off the expression of the hyperactive genes and then observed how this affected sarcoma growth. Hettmer et al. found that blocking the expression of one particular gene, which encodes an enzyme called asparagine synthetase, slowed down the growth of the sarcoma the most.

Asparagine synthetase makes the amino acid asparagine, which is needed to make proteins in cells. Further experiments showed that reducing the amount of asparagine in human and mouse sarcoma cells slowed down the growth of these cells. A drug that lowers the amount of asparagine in cells is already used to treat some blood cancers. Hettmer et al.'s findings suggest that drugs that alter the availability of asparagine in the body might also be useful to treat sarcomas with mutant forms of Ras.

oncogenic mutations or exclusive chromosomal translocations within a relatively simple karyotype (*Bovee and Hogendoorn, 2010*). For RMS, two main genotypes have been described: those characterized by expression of the fusion oncogenes *PAX3:FOXO1* or *PAX7:FOXO1* and those that lack these fusions. The most common oncogenic mutations in the latter group of fusion-negative RMS tumors are in the Ras pathway (*Shern et al., 2014*; *Chen et al., 2013*).

We previously reported rapid sarcoma induction by intramuscular implantation of *Kras (G12V)*-expressing, *Cdkn2a (p16p19)* deficient mouse myofiber-associated (MFA) cells into the extremity muscles of NOD. SCID mice (*Hettmer et al., 2011*). Transcriptional profiling of *Kras; p16p19$^{null}$* sarcomas identified a cluster of sarcoma-relevant candidate genes. These genes are enriched in mouse sarcomas and in human RMS as compared to normal mouse or human skeletal muscle (*Hettmer et al., 2011*), and may include transcripts of fundamental importance for sarcoma growth. To examine the contributions of each of these candidate genes to sarcoma growth, we performed a customized shRNA-based proliferation screen. The strongest inhibitory effect on sarcoma cell proliferation was observed after silencing of asparagine synthetase (ASNS), the enzyme that catalyzes cellular synthesis of the non-essential amino acid asparagine. We found that adequate availability of asparagine is required in rapidly proliferating sarcomas cells, likely to support nascent polypeptide synthesis, and that asparagine starvation impedes sarcoma growth. Thus, small molecules targeting asparagine availability (*Richards and Kilberg, 2006*) could be useful as anti-sarcoma therapeutics.

## Results

### *Kras;p16p19$^{null}$* mouse sarcomas identify a cluster of sarcoma-relevant genes

In prior work, we showed that ex-vivo lentiviral transduction with oncogenic *Kras (G12v)* of *Cdkn2a* (p16p19)-deficient myofiber-associated (MFA) cells, isolated by fluorescence activated cell sorting (FACS) from muscle tissue of *Cdkn2a$^{-/-}$* mice, drives rapid sarcoma formation upon transplantation of these cells into the muscle of immunocompromised mice (*Hettmer et al., 2011*) (*Figure 1—figure*

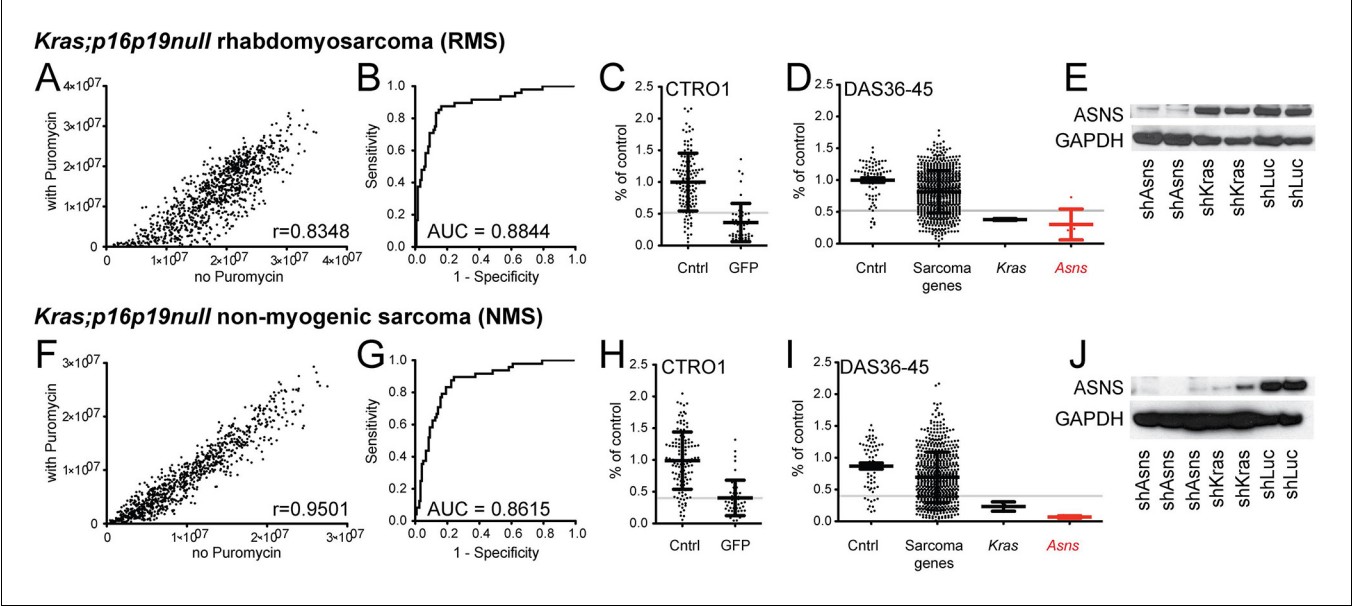

**Figure 1.** Functional genomic screening identified asparagine synthetase (ASNS) as a high-priority sarcoma target. 141 sarcoma-relevant genes were identified by prior transcriptional profiling of genetically engineered *Kras;p16p19$^{null}$* mouse rhabdomyosarcomas (RMS) and non-myogenic sarcomas (NMS) (*Hettmer et al., 2011*). (A–D, F–I) The contributions of each of the 141 sarcoma-relevant genes to sarcoma cell proliferation were determined by customized shRNA screening. (B–D, G–I). The screen contained a control set, including cells exposed to shLUC, shRFP, shLACZ (cntrl; predicted to have no effect on cell proliferation) and cells exposed to shGFP (GFP; predicted to silence Kras (G12V)-IRES-GFP and reduce cell proliferation). (B,G) Receiver operator curve analysis using cntrl-shRNA-infected cells as negative and shGFP-infected cells as positive controls determined a false discovery rate of <30% for shRNAs associated with a reduction in proliferation to <52% of the average of cntrl-shRNA-infected RMS cells (grey line in panel C) and to <40% of cntrl-shRNA-infected NMS cells (grey line in panel H). (D, I) The shRNA screen included cells exposed to shLUC, shRFP, shLACZ (cntrl), shKRAS and shRNAs directed against each of the 141 candidate genes (5 shRNAs per gene). ShRNAs directed against the gene encoding Asparagine Synthetase (*Asns*) showed the strongest effect on NMS and RMS proliferation (p<0.0001, q<0.01, 4–5 of 5 shRNAs with FDR<30%). (E–J) Effective ASNS knockdown by the shRNAs used in the screen was confirmed by Western blot. See **Supplementary files 1–4** for raw data from the shRNA screen, and **Supplementary files 5–6** for scores for each of the 141 candidate genes. Significance levels were defined as follows: 1, 5 shRNAs with FDR<30%; 2, 4 shRNAs with FDR<30%; 3, 3 shRNAs with FDR<30%.

The following figure supplements are available for Figure 1:

**Figure supplement 1.** Sarcoma induction strategy.

supplement 1). The myogenic differentiation status of the *Ras*-driven sarcomas generated in this system depends largely on the cell type transduced, also known as the "cell-of-origin": *Kras; p16p19$^{null}$* satellite cells typically gave rise to RMS, whereas the identical oncogenetic lesions introduced into fibroadipogenic precursors within the MFA cell pool almost always produced sarcomas lacking myogenic differentiation features (non-myogenic sarcomas, NMS) (*Hettmer et al., 2011*) (*Figure 1—figure supplement 1*). We previously showed that *Kras;p16p19$^{null}$* mouse sarcomas from each of these cellular origins recapitulate transcriptional profiles across the entire spectrum of human RMS and used this information to identify 141 genes of potential significance for sarcoma growth (*Hettmer et al., 2011*). To evaluate the functional contributions of each of the previously identified sarcoma-relevant genes, we designed a customized shRNA proliferation screen, using 5 distinct shRNAs per candidate gene (*Supplementary files 1–4*). The screen was carried out in one *Kras; p16p19$^{null}$* RMS and one *Kras;p16p19$^{null}$* NMS cell line, and shRNAs were delivered in puromycin-selectable pLKO lentiviral vectors. Correlation coefficients of 0.8348 and 0.9501 between puromycin-treated and untreated cells confirmed adequate transduction efficiencies (*Figure 1A,F*). As shRNA mediated silencing of Kras (G12v)-IRES-GFP, the driver oncogene used to initiate the mouse sarcomas, markedly inhibits the growth of *Kras;p16p19$^{null}$* sarcoma cells (*Figure 2A–B*, *Figure 2—figure supplement 1A–B*), shRNAs directed against either GFP or KRAS served as positive controls in this screen and showed clear growth-inhibitory effects (*Figure 1C–D, 1H-I*). Control shRNAs (cntrl-

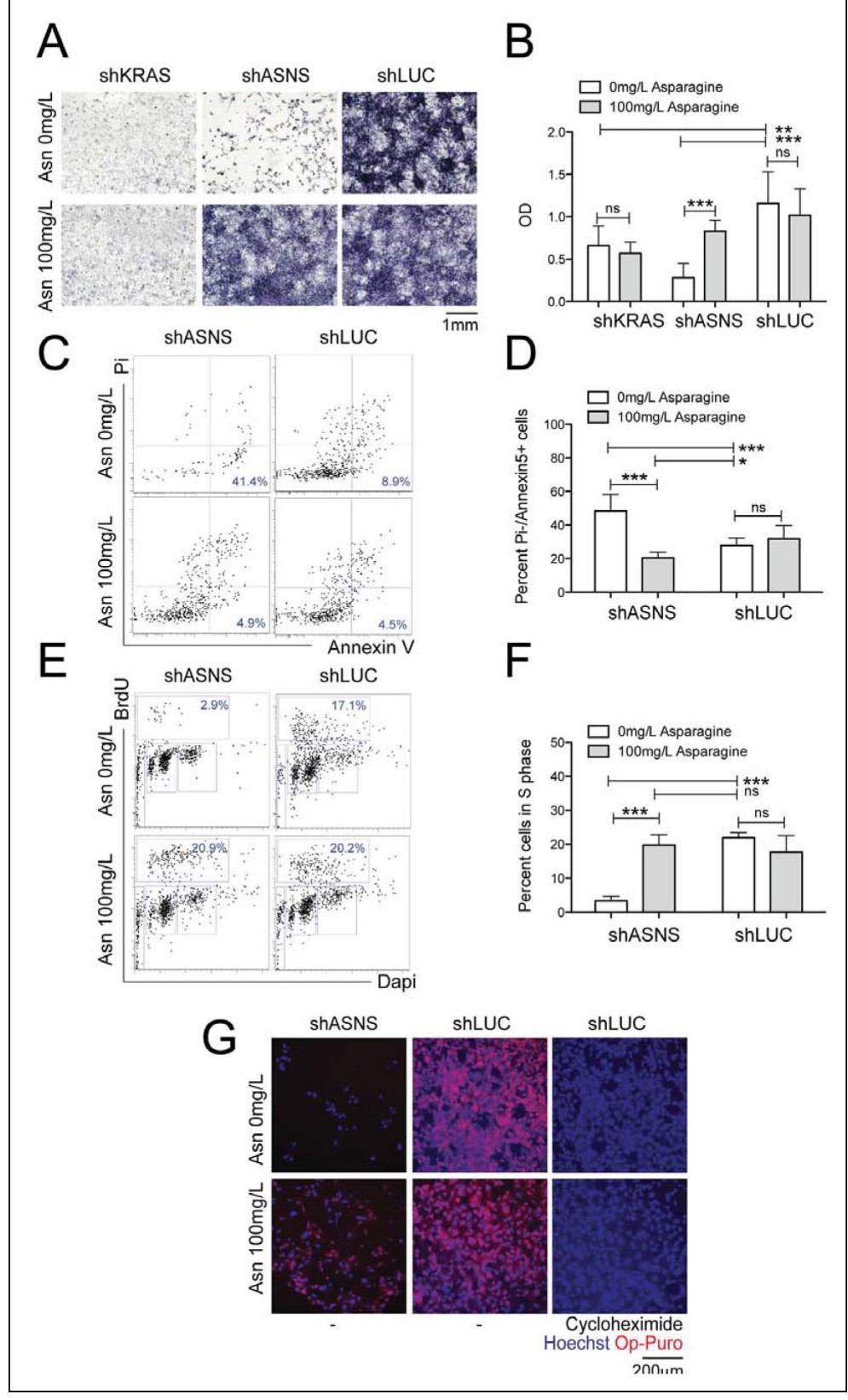

**Figure 2.** Reduced growth of mouse *Kras;p16p19*$^{null}$ RMS cells after *Asns* silencing is associated with inhibition of polypeptide synthesis. (**A–B**) ShRNA-mediated silencing of *Asns* and *Kras* in a mouse *Kras;p16p19*$^{null}$ RMS cell line reduced proliferation activity compared to shLUC-infected control cells as measured by MTT uptake. Asparagine

*Figure 2. continued on next page*

*Figure 2. Continued*

supplementation (100 mg/L) in the tissue culture medium reversed the anti-proliferative effects of shASNS but not shKRAS. (**C–F**) *Asns* silencing increased the (**C–D**) percentage of apoptotoc (PI-/Annexin5+) cells and reduced the (**E–F**) percentage of S phase cells as determined by BrdU staining, compared to shLUC-infected control cells. Both effects were reversed by exogenous Asparagine supplementation (100 mg/L). (**G**) Polypeptide synthetic activity was determined by OP-puromycin staining. Absent OP-puromycin staining in cells treated with cycloheximide (right panels), an inhibitor of protein translation, validated the experimental approach. *Asns* silencing reduced polypeptide synthesis in *Kras;p16p19$^{null}$* RMS cells (top left panel), and polypeptide synthesis was restored in shASNS RMS cells by Asparagine supplementation (bottom left panel). (**A–F**) Data were evaluated for statistical significance by T-tests (ns p$\geq$0.05, *p<0.05, **p<0.01, ***p <0.001). See *Figure 2—figure supplement 1* for similar effects of Asns silencing in mouse *Kras;p16p19$^{null}$* NMS cells.

The following figure supplements are available for Figure 2:

**Figure supplement 1.** Reduced mouse *Kras;p16p19$^{null}$* NMS cell growth after *Asns* silencing was associated with reduced polypeptide synthesis.

**Figure supplement 2.** Asparagine concentrations of 10 or 100 mg/L reverse the effects of ASNS silencing on sarcoma growth.

**Figure supplement 3.** Glutaminase expression in mouse *Kras;p16p19$^{null}$* sarcoma cells.

shRNA) directed against LACZ, red fluorescent protein (RFP) and luciferase (LUC) served as negative controls, and showed no growth-inhibitory effects (*Figure 1C–D, 1H-I*). Receiver operator curve (ROC) analysis, using an external control set of shGFP-infected and cntrl-shRNA-infected RMS and NMS cells (*Supplementary files 3–4*), validated the ability of this system to distinguish between shRNAs with and without growth-inhibitory effects on sarcoma cell proliferation (*Figure 1B,G*). ROC analysis also determined a false discovery rate of <30% for shRNAs associated with a reduction in proliferation to <52% of the average of cntrl-shRNA-infected RMS cells (*Figure 1B–C*) and <40% of cntrl-shRNA-infected NMS cells (*Figure 1G–H*). In both RMS and NMS cells, silencing of ASNS (Asparagine Synthetase) produced by far the strongest anti-proliferative effect (p<0.0001, q<0.01, 4–5 of 5 shRNAs with FDR<30%; *Figure 1D,I* and *Supplementary files 5–6*). ASNS silencing reduced the growth of *Kras;p16p19$^{null}$* RMS and NMS cells to 30.16% and 6.69% of the average of control RMS and NMS cells, respectively. Effective depletion of ASNS protein by the target-specific shRNAs employed in our screen was confirmed by Western Blot (*Figure 1E,J*).

## ASNS silencing inhibits growth of mouse *Kras;p16p19$^{null}$* sarcoma cells by asparagine starvation

ASNS encodes the enzyme asparagine synthetase, which converts aspartate into asparagine using glutamine as a nitrogen donor. Therefore, we next tested if the growth-inhibitory effects of genetic ASNS inhibition (*Figure 1D,I*) on mouse sarcoma growth were reversed by exogenous supplementation with the ASNS product asparagine. Supplementation of the culture media with 100 mg/L asparagine reversed the growth inhibition observed in shASNS-infected *Kras;p16p19$^{null}$* RMS (*Figure 2A-2B*) and NMS (*Figure 2—figure supplement 1A-1B*) cells. Dose response analysis revealed that reversal of growth inhibiton of shASNS-transduced *Kras;p16p19$^{null}$* RMS and NMS cells was also observed at 10 mg/L asparagine, whereas 0.01, 0.1 or 1 mg/L asparagine were insufficient (*Figure 2—figure supplement 2*). Asparagine supplementation did not reverse the growth inhibitory effect of shRNA-mediated silencing of *Kras* in *Kras;p16p19$^{null}$* RMS cells (*Figure 2A-2B*). Taken together, we conclude that the growth-inhibitory effects of ASNS silencing on mouse sarcoma cells result from cellular asparagine starvation.

## Asparagine starvation reduces cell proliferation, increases cell death and impedes nascent polypeptide synthesis in mouse *Kras; p16p19$^{null}$* sarcoma cells

*Asns* silencing in mouse *Kras;p16p19$^{null}$* RMS cells caused an increase in the fraction of sarcoma cells undergoing apoptosis, as determined by staining with propidium iodide (PI) and Annexin V

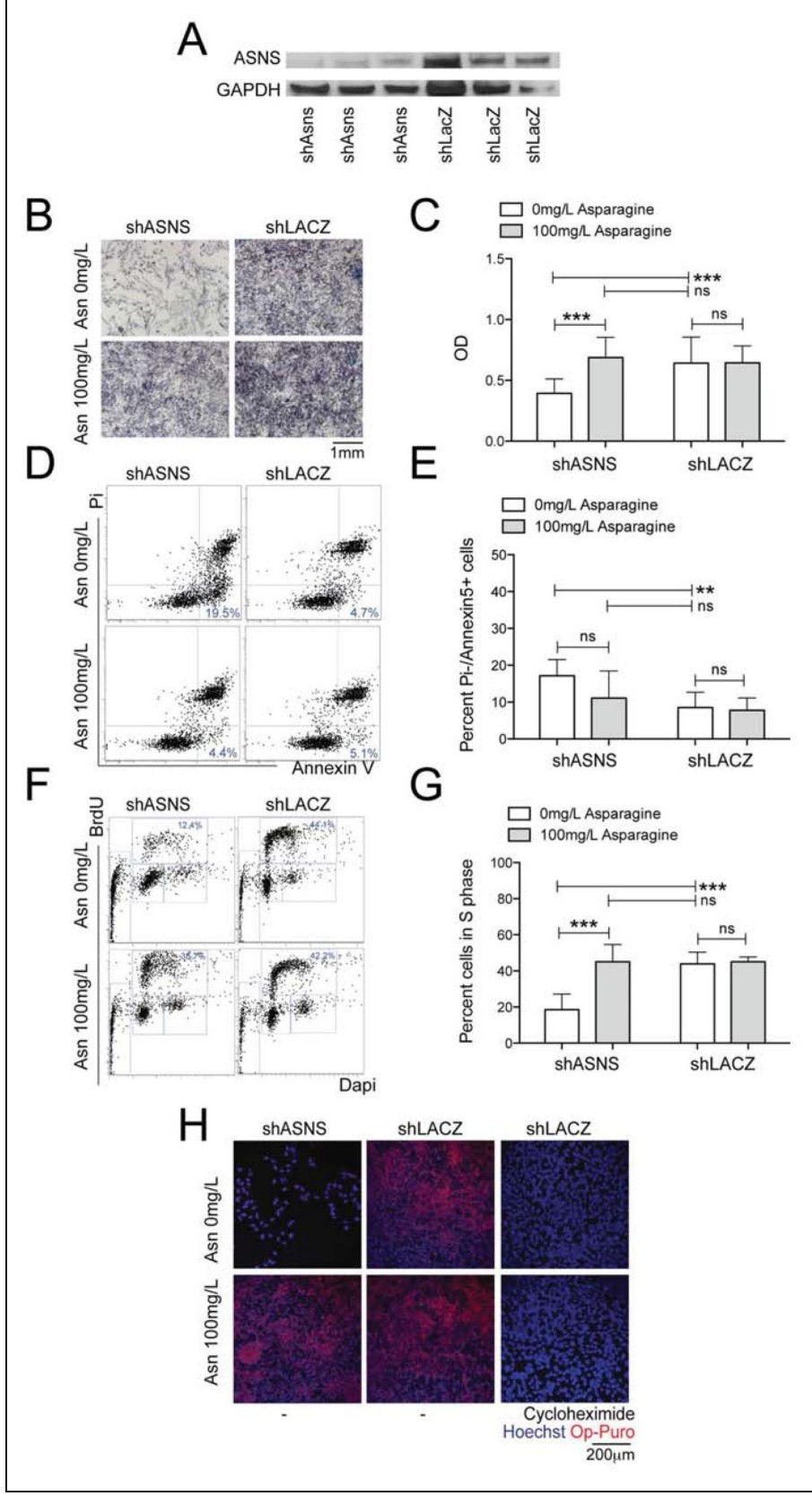

**Figure 3.** Reduced growth of human Rh30 RMS cells after *ASNS* silencing is associated with reduced polypeptide synthesis. (**A**) ShRNA-mediated silencing of *ASNS* in Rh30 cells was validated by Western Blot. (**B–C**) ASNS silencing reduced proliferation compared to shLACZ-infected control cells
*Figure 3. continued on next page*

Figure 3. Continued

as measured by MTT uptake. Asparagine supplementation (100 mg/L) in the tissue culture medium reversed the anti-proliferative effects of shASNS. (D–G) *ASNS* silencing increased the (D–E) percentage of apoptotic (PI-/Annexin5+ ) cells and reduced the (F–G) percentage of S-phase cells , as compared to shLACZ-infected control cells. (F–G) Exogenous Asparagine supplementation reversed shASNS effects on cell cycle progression. (H) Polypeptide synthetic activity was determined by OP-puromycin staining. Absent OP-puromycin staining in Rh30 cells treated with cycloheximide (right panels) validated the experimental approach. *ASNS* silencing reduced polypeptide synthesis (top left panel), and polypeptide synthesis was restored in shASNS RMS cells by Asparagine supplementation (bottom left panel). (B–G) Data were evaluated for statistical significance by T-tests (ns p$\geq$0.05, * p<0.05, ** p<0.01, *** p <0.001).

(p<0.001, *Figure 2C-2D*). Moreover, *Asns* silencing reduced the percentage of BrdU + cells in S phase (p<0.001, *Figure 2E-2F*). Both effects were reversed by exogenous asparagine supplementation (*Figure 2D,F*). To evaluate whether cellular asparagine starvation due to *Asns* silencing impedes sarcoma cell proliferation by interfering with the cells' ability to generate nascent polypeptide chains, *Kras;p16p19$^{null}$* RMS cells were exposed to O-propargyl-puromycin (OP-puromycin). OP-puromycin forms covalent conjugates with newly synthesized polypeptides, which can be visualized by azide-alkyne cycloaddition. Rapidly proliferating shLUC-infected *Kras;p16p19$^{null}$* RMS cells exhibited strong OP-puromycin staining, indicating brisk polypeptide synthesis (*Figure 2G*, middle panels). However, blockade of protein translation by exposure to cycloheximide abrogated OP-puromycin staining (*Figure 2G*, far right panels). Similarly, shASNS-infected cells exhibited only minimal OP-puromycin staining (*Figure 2G*, upper left panel), while synthesis of new polypeptides was restored in shASNS-infected sarcoma cells grown in medium supplemented with asparagine (*Figure 2G*, lower left panel). Similar effects of ASNS silencing on apoptosis, cell cycle and synthesis of nascent peptide chains were observed in *Kras;p16p19$^{null}$* NMS cells (*Figure 2—figure supplement 1*).

## Asparagine starvation impedes human RMS growth and polypeptide synthesis

ASNS expression was evaluated in primary human sarcoma tissue by immunohistochemistry (IHC) using a commercially available tissue array (US Biomax SO2081). ASNS was detected in 16 of 22 (73%) human RMS cores (*Figure 4—figure supplement 1A*) and in 12 of 27 (44%) human leiomyosarcoma cores (*Figure 4—figure supplement 1B*). Also, increased expression of *ASNS* compared to normal human muscle was detected in 9 of 9 human sarcoma cell lines analyzed by PCR (*Figure 4—figure supplement 1C*), including the PAX3:FOXO1-positive human RMS cell line Rh30. To evaluate the impact of *ASNS* silencing on human RMS cells, we transduced Rh30 cells with lentiviruses encoding shASNS or control (shLACZ) shRNAs (*Figure 3*). ShRNA-mediated knockdown of *ASNS* in Rh30 cells (*Figure 3A*) reduced proliferation (p<0.001; *Figure 3B–C*), increased the percentage of apoptotic cells (p<0.01; *Figure 3D–E*), reduced the percentage of cells in S phase (p<0.001; *Figure 3F–G*) and impeded nascent polypeptide synthesis (*Figure 3H*). The effects of *ASNS* silencing on Rh30 growth and peptide synthesis were reversed by asparagine supplementation (*Figure 3B–H*). Thus, ASNS silencing in human Rh30 cells recapitulated the inhibitory effects on cell growth and polypeptide synthesis observed in mouse *Kras;p16p19$^{null}$* sarcoma cells.

## Chemical targeting of Asparagine availability reduces sarcoma growth

Asparagine homeostasis represents an actionable cellular process. Amino sulfoximines directly inhibit ASNS activity (*Ikeuchi et al., 2012*; *Richards and Kilberg, 2006*), whereas asparaginase, an FDA-approved drug widely used in the treatment of leukemia, hydrolyzes asparagine to aspartate and ammonia. Both amino sulfoximine 5 (AS5) and asparaginase reduced the proliferation of mouse and human sarcoma cell lines in vitro (*Figure 4A–C*). For asparaginase, EC50 concentrations were estimated at 0.2–0.5 IU/ml in mouse *Kras;p16p19$^{null}$* sarcoma cells, 0.8–0.9 IU/ml in human HT1080, Rh30 and Rh41 cells and 6 IU/ml in human RD cells (*Figure 4A–B*). For the ASNS inhibitor AS5, EC50 concentrations were estimated at 80–150 µM in mouse *Kras;p16p19$^{null}$* sarcoma cells and 200–300 µM in the human sarcoma cell lines tested (*Figure 4A,C*). Similar to previous observations in cells that underwent genetic inhibition of ASNS, the growth-inhibitory effects of chemical ASNS inhibition by AS5 were reversed by exogenous asparagine supplementation (*Figure 4D*). These findings

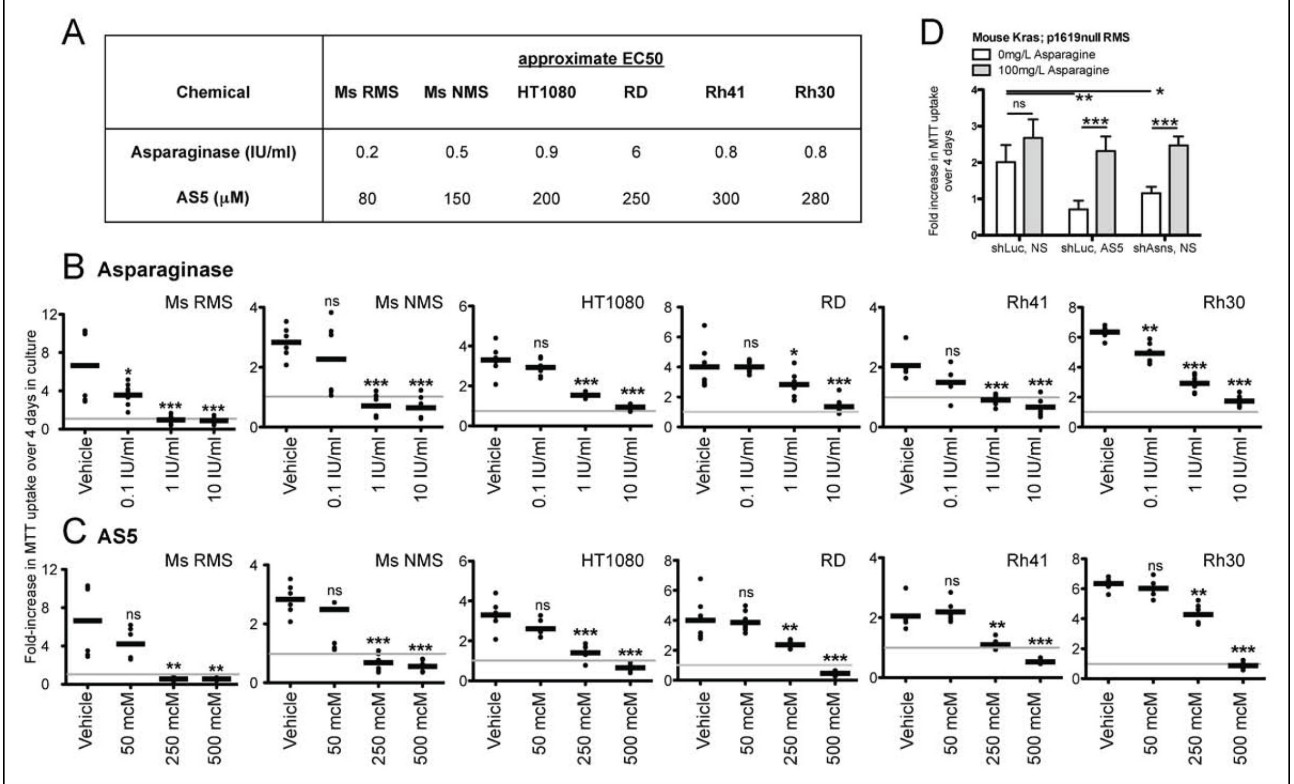

**Figure 4.** Inhibition of mouse and human sarcoma cell growth in vitro by chemical compounds interfering with Asparagine homeostasis. (A–C) Proliferation assays of mouse (Ms RMS, Ms NMS) and human (HT1080, RD, Rh41, Rh30) sarcoma cell lines exposed to the indicated doses of chemical modulators of Asparagine homeostasis: (A–B) Asparaginase or (A,C) AS5. Both chemicals were diluted in 0.9% NaCl (Normal Saline (NS)) as vehicle. (D) Chemical and genetic ASNS inhibition was reversed by exogenous supplementation with 100 mg/L asparagine in the tissue culture medium (100 mg/L, which corresponds to 757 μM; compared to normal asparagine concentrations in mouse and human plasma of 29 μM and 55 μM, respectively (*Cooney et al., 1970*)). Data were evaluated for statistical significance by T-tests (ns $p \geq 0.05$, * $p < 0.05$, ** $p < 0.01$, *** $p < 0.001$). See *Figure 4—figure supplement 1* for ASNS expression in human sarcoma cells.

The following figure supplements are available for Figure 4:

**Figure supplement 1.** Expression of candidate sarcoma targets in human sarcoma tissue.

strongly suggest that the growth inhibitory effects of AS5 result from asparagine deprivation of sarcoma cells.

## Asparagine depletion impedes mouse sarcoma growth in vivo

Due to the poor cell permeability of AS5, its growth-inhibitory effects on sarcoma cells required EC50 concentrations greater than 80 μM, making in vivo testing infeasible. Thus, to determine the effects of reduced ASNS activity on sarcoma growth in vivo, 100 shASNS-infected and 100 shLUC-infected *Kras; p16p19^null* RMS cells were implanted into the cardiotoxin-preinjured gastrocnemius muscles of 1- to 3-months old NOD. SCID mice (*Figure 5*). IHC showed that tumors arising from shASNS cells expressed less ASNS than tumors arising from shLUC cells (*Figure 5A*). However, there was no difference in latency of shASNS- and shLUC-tumors (p = 0.3; *Figure 5B*).

Our in vitro data suggested that growth inhibition induced by ASNS silencing can be rescued by provision of exogenous aparagine at concentrations between 1 and 10 mg/L (*Figure 2—figure supplement 2*). As normal asparagine concentrations in mouse and human plasma were previously reported to be between 3.8 mg/L and 7.3 mg/L (*Cooney et al., 1970*), these data suggest that freely available asparagine in mouse serum and tissue might counteract the effects of tumor-specific ASNS silencing. To examine this possibility, we treated subgroups of animals transplanted with shASNS- or shLUC-tumor cells with asparaginase (1500 IU/kg; (*Szymanska et al., 2012*)) by

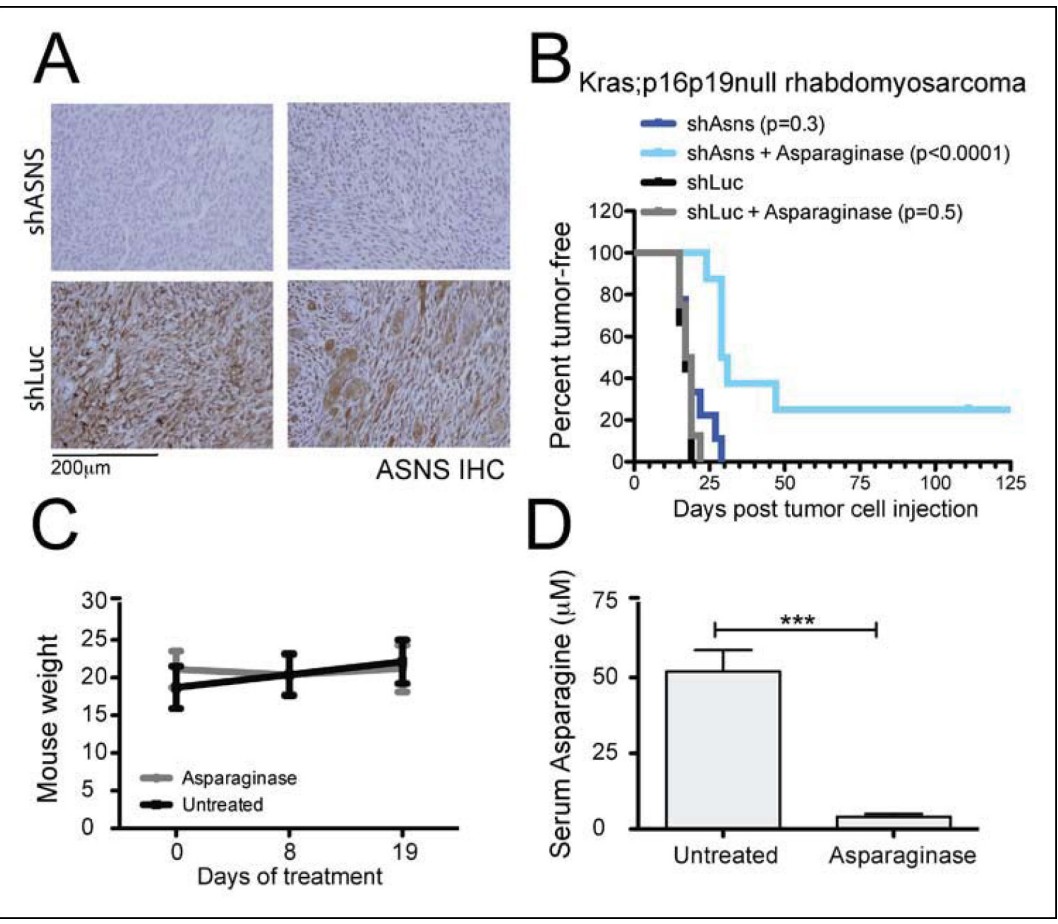

**Figure 5.** *Asns* silencing delayed *Kras;p16p19^null^* RMS growth in Asparagine-depleted mice. (**A**) Tumors arising from shASNS RMS cells expressed less ASNS than tumors arising from shLUC cells as shown by IHC staining. Of 6 tumors arising from shASNS cells, cytoplasmic ASNS positivity was observed in 50–75% of cells in one tumor, in 25–50% of cells in 3 tumors and in <25% of cells in 2 tumors. Of 5 tumors arising from control shLUC-cells, cytoplasmic ASNS staining was detected in >75% of cells in 3 tumors and in 25–50% of cells in 2 tumors. Representative images are shown. (**B**) Effects of *Asns* silencing on tumor growth in vivo were evaluated by transplantation. One subgroup of recipient mice was treated with Asparaginase (Elspar) by daily intraperitoneal (IP) injections at a dose of 1500 IU/kg. ShASNS silencing delayed tumor onset in recipient mice treated with Asparaginase compared to shLUC-infected RMS cells (p<0.0001). (**C**) Asparaginase-treated mice maintained their weight over the course of a 19-day exposure. Each experimental group included 5 mice, and findings were replicated in 2 independent transplantation experiments. (**D**) Daily IP injection of Asparaginase results in a 13-fold reduction in serum Asparagine levels from 52 ± 7 to 4 ± 1 μmol/L. Differences in tumor latency were evaluated for statistical significance by logrank (Mantel-Cox) test. Differences in serum amino acid concentrations were determined by T-test (ns p≥0.05, *p<0.05, **p<0.01, ***p <0.001). See *Figure 5—figure supplement 1* for similar effects of *Asns* silencing in *Kras;p16p19^null^* NMS cells on secondary tumor induction and *Supplementary file 7* for changes in serum amino acid levels in Asparaginase-treated versus control mice.

The following figure supplements are available for Figure 5:

**Figure supplement 1.** *Asns* silencing delayed *Kras;p16p19^null^* NMS growth in Asparagine-depleted mice.

daily intraperitoneal (IP) injection. Asparaginase treatment was initiated on the day of tumor cell injection and continued for 35–41 days. This dosage was well tolerated by the animals without significant weight loss (*Figure 5C*). Serum asparagine levels were reduced 13-fold in asparaginase-treated mice (0.53 mg/L (4 μM) versus 6.87 mg/L (52 μM) in untreated mice, p<0.001; *Figure 5D* and *Supplementary file 7*). Daily exposure to asparaginase did not change the latency of shLUC-tumors (p = 0.5; *Figure 5B*); however, asparaginase treatment significantly prolonged tumor latency in mice

implanted with shASNS-RMS cells (p<0.001; *Figure 5B*). Moreover, 2 out of 8 mice in this experimental group did not develop tumors during 4 months of follow up after tumor cell injection. Similar effects were observed when NOD. SCID mice were transplanted with shASNS-infected and shLUC-infected *Kras; p16p19^null* NMS cells (*Figure 5—figure supplement 1*). Thus, ASNS inhibition combined with depletion of plasma asparagine reduces sarcoma growth in vivo.

## Discussion

Cancer cells depend on biological mechanisms that guarantee adequate provision of energy and biosynthetic precursors to support cell growth. Functional genomic screening of genetically engineered mouse sarcomas revealed that asparagine synthetase (ASNS) exerted the strongest observed effect on sarcoma cell proliferation within a small group of genes upregulated in both mouse and human sarcomas. ASNS is the amidotransferase that converts L-aspartic acid into L-asparagine in an energy-consuming enzymatic reaction requiring ATP and a nitrogen source that is L-glutamine in eukaryotic cells (*Richards and Kilberg, 2006*). Depletion of functional ASNS in both mouse and human RMS cells reduced the proportion of cells in S phase and impeded synthesis of nascent polypeptide chains. Similarly, ASNS silencing in melanoma cells was recently reported to result in cell cycle arrest (*Li et al., 2015*). The effects of ASNS reduction in sarcoma cells were reversed by exogenous supplementation with asparagine, supporting the notion that rapidly growing sarcoma cells depend on adequate availability of intrinsic or extrinsic asparagine to support tumor growth. Moreover, ASNS inhibition significantly slowed mouse sarcoma growth in vivo only when combined with depletion of plasma asparagine, likely reflecting the ability of systemic asparagine in the tumor environment to replenish intracellular asparagine availability after ASNS inhibition. We speculate that asparagine reliance of sarcoma cells represents a metabolic vulnerability that could be exploited therapeutically to inhibit rapid tumor growth.

Previously published metabolic profiling studies have identified a number of metabolites that are heavily consumed by cancer cell lines, including glycine and asparagine (*Jain et al., 2012*). Glycine starvation prolonged the G1 phase of the cell cycle and reduced proliferation, in part because sufficient amounts of this amino acid are required to support de novo purine biosynthesis in rapidly dividing cells. Unlike purine synthesis, protein synthesis in glycine-starved cells remained relatively intact (*Jain et al., 2012*). In contrast, we found that asparagine starvation of mouse and human sarcoma cells impedes synthesis of nascent polypeptide chains thereby slowing cell proliferation. These results are consistent with observations in soybean, barley and maize where free asparagine levels in developing seeds correlate positively with higher protein levels at maturity (*Pandurangan et al., 2012*). Moreover, limiting the extracellular supply or blocking the synthesis of single amino acids is known to suppress global translation initiation via activation of GCN2 and phosphorylation of the translation initiation factor EIF2α (*Sood et al., 2000*). Thus, asparagine in the tumor cell environment may have important functions in controlling the synthesis and turnover of protein in sarcoma cells.

Impaired peptide biosynthesis may not be the only mechanism contributing to the asparagine dependence of sarcoma cells described in this study. For instance, our data do not exclude the possibility that glutamate deprivation and/or aspartate excess resulting from changes in asparagine biosynthesis negatively impact cell survival and proliferation. However, aspartate carries reducing equivalents in the malate-aspartate shuttle (*Son et al., 2013*), and increased aspartate levels after ASNS knockdown might be predicted to benefit the redox state of sarcoma cells. In addition, it has been suggested that increased glutaminase expression in cancer cells can counterbalance decreased glutamate levels, especially in a glutamine-rich environment (*Huang et al., 2014*). Consistent with this notion, we found that glutaminase expression was increased by 7- to 11-fold in mouse sarcoma cells compared to normal mouse muscle (*Figure 2—figure supplement 3*), suggesting that this mechanism may be used by sarcoma cells to counteract glutamate reductions that may result from ASNS silencing.

Finally, a recent report demonstrated, unexpectedly, that asparagine supplementation can suppress cell death in glutamine-deprived cells (*Zhang et al., 2014*), implicating asparagine in promoting cellular adaptation to metabolic stresses such as glutamine depletion. This study also noted that asparagine is the last amino acid synthesized in the TCA cycle and that its amination depends exclusively on glutamine (*Zhang et al., 2014*). Amino acid availability is known to stimulate mechanistic target of rapamycin (mTOR) complex 1, which integrates environmental and intracellular signals to

regulate cell growth (*Jewell et al., 2015*). Taken together, these observations suggest that asparagine may serve a central role as a cellular sensor of TCA cycle intermediate/ reduced nitrogen availability and, ultimately, as a metabolic regulator of cell behavior.

While our studies demonstrate a clear dependence of sarcoma cell growth and survival on cellular asparagine levels, they certainly do not exclude the possibility that adequate availability of amino acids other than asparagine may also be important. The shRNA proliferation screen we performed was designed to evaluate the functional contributions of a particular group of sarcoma signature genes identified by their high level expression in both *Ras*-driven mouse sarcomas and human sarcomas, and thus did not comprehensively evaluate all amino acid biosynthetic pathways. Indeed, within the cluster of sarcoma genes evaluated, the top-scoring cellular functions were cell cycle control and cell division (*Hettmer et al., 2011*). However, we note that in addition to ASNS, our list of candidate targets included genes encoding 3 other enzymes relevant for amino acid biosynthesis: branched chain aminotransferase 1 (BCAT1), phosphoserine aminotransferase (PSAT1) and phosphoglycerate dehydrogenase (PHGDH). Both PHGDH and PSAT1 contribute to serine/ glycine biosynthesis, which plays an important role in supporting nucleotide synthesis in rapidly proliferating cells (*Jain et al., 2012*; *Labuschagne et al., 2014*), and recent publications indicate that high expression of PHGDH in tumor tissue is required for cell growth in epithelial malignancies (*Locasale et al., 2011*; *Possemato et al., 2011*). However, in our screen, silencing of PSAT1, PHGDH or BCAT1 did not inhibit sarcoma cell growth (*Supplementary files 5–6*), suggesting that these biosynthetic pathways may be of lesser importance for the sarcomatous malignancies studied here.

ASNS expression among tissues in adult animals varies considerably (*Balasubramanian et al., 2013*). ASNS in tumor tissue has been linked to the transactivating effects of oncogenic effectors such as *TP53* (*Scian et al., 2004*) and metabolic stress (*Balasubramanian et al., 2013*; *Cui et al., 2007*). For example, in pancreatic cancer, glucose deprivation upregulated ASNS expression, which, in turn, protected tumor cells from apoptosis induced by glucose deprivation itself (*Cui et al., 2007*). Similarly, upregulation of ASNS in response to amino acid restriction, such as plasma asparagine depletion by treatment with asparaginase (*Cui et al., 2007*), is part of a normal physiological adaptation response to counteract nutrient deprivation (*Balasubramanian et al., 2013*). As sarcomas outgrow the existing vasculature, tumor cells are continuously exposed to a microenvironment in which the supply of nutrients is limited. Increased *Asns* mRNA expression in *Kras;p16p19^{null}* mouse sarcomas as compared to normal muscle could occur in response to amino acid and glucose deprivation in rapidly growing sarcomas.

Cellular asparagine reliance has been exploited successfully in the treatment of acute lymphoblastic leukemia (ALL) with bacterially derived asparaginase (*Haskell et al., 1969*; *Jaffe et al., 1971*; *Richards and Kilberg, 2006*). Lymphoblasts are thought to be exquisitely sensitive to asparaginase treatment due to their low ASNS expression (*Richards and Kilberg, 2006*). Growth inhibitory effects of asparaginase on sarcoma cells in vitro were previously reported by Tardito et al (*Tardito et al., 2006*). However, the response of sarcoma cells to asparaginase alone is moderate to poor (*Figure 4A*, (*Tardito et al., 2006*)) when compared to the published spectrum of asparaginase sensitivity of primary lymphoblasts (EC50 concentrations between <0.002 and >10 IU/ml; (*Fine et al., 2005*)). One published report on the efficacy of asparaginase as a single agent documented remissions in 7 of 32 children with ALL, but there was no objective response in a single patient with RMS (*Jaffe et al., 1971*). Yet, asparaginase is a well-characterized drug with a relatively favorable toxicity profile that does not overlap with the toxicities of conventional cytostatics used in RMS treatment (*Haskell et al., 1969*), and the further development of specific, cell-permeable chemical inhibitors of ASNS may open additional therapeutic opportunities (*Richards and Kilberg, 2006*), especially when combined with systemic asparagine depletion.

Understanding the molecular networks that support sarcoma cell proliferation may enable the development of therapies based on selective targeting of proliferation-relevant cellular pathways. This study identified asparagine starvation as a candidate intervention that impedes sarcoma growth. Yet, it is highly unlikely that any interventions will have noticeable anti-sarcoma effects in vivo when used alone. For future studies and clinical development, it will be important to rationally select combinations of interventions that target multiple proliferation-relevant cellular processes including Asparagine reliance of sarcoma cells.

## Materials and methods

### *Kras;p16p19*$^{null}$ mouse sarcomas

Primary *Kras;p16p19*$^{null}$ mouse sarcomas were induced by ex-vivo transduction of freshly sorted Cdkn2a$^{-/-}$ (*p16p19*$^{null}$) mouse skeletal muscle precursor cells (CD45$^-$CD11b$^-$TER119$^-$Sca1$^-$CXCR4$^+$ β1integrin$^+$) or Sca1+ fibroadipogenic precursor cells CD45$^-$CD11b$^-$TER119$^-$Sca1$^+$) with pGIPZ-Kras (G12V)-IRES-GFP lentivirus followed by intramuscular transplantation of *Kras*-expressing, *p16p19*$^{null}$ cells into the cardiotoxin-preinjured gastrocnemius muscles of 1- to 3-months old NOD/ SCID mice, as previously described (*Hettmer et al., 2011*). Secondary *Kras;p16p19*$^{null}$ mouse sarcomas were generated by implanting 100 *Kras;p16p19*$^{null}$ mouse RMS or NMS cells into the cardiotoxin-preinjured gastrocnemius muscles of 1- to 3-months old NOD/SCID mice.

### Human skeletal muscle

Use of human muscle was approved by the Institutional Review Board at Joslin Diabetes Center. Human fetal muscle was obtained from 20–23 week gestation fetuses and adult muscle from deceased volunteers. Tissue was homogenized in TRIzol using a tissue homogenizer prior to RNA isolation.

### Mice

C57BL6, NOD/CB17-Prkdcscid/J (NOD/SCID) and *p16p19*$^{null}$ mice (B6.129 background) mice were obtained from Jackson Laboratory and the National Institutes of Health/Mouse Models of Human Cancer Consortium, respectively. Mice were bred and maintained at the Joslin Diabetes Center Animal Facility. All animal experiments were approved by the Joslin Diabetes Center Institutional Animal Care and Use Committee.

### Sarcoma cell lines

Mouse sarcoma cell lines were established from 2 *Kras;p16p19*$^{null}$ mouse RMS tumors (T14-R, SMP-01) and one *Kras;p16p19*$^{null}$ mouse NMS tumor (Sca1-01). The human RMS cell line RD (translocation-negative) and the human fibrosarcoma line HT1080 were purchased from ATCC. Human RMS cell lines Rh3, Rh5, Rh10, Rh28, Rh30, Rh41 (all PAX3:FOXO1-positive) and Rh36 (translocation-negative) were gifts from Dr. Peter Houghton (Nationwide Children's Hospital, Columbus, OH). All cell lines were maintained in DMEM supplemented with 10% FBS and 1% Penicillin-Streptomycin. To evaluate the effects of asparagine supplementation on sarcoma cells in vitro, 4.15 g DMEM (D5030, Sigma, St. Louis, MO), 2.25 g Glucose (Sigma), 1.85 g NaHCO$^3$ (Sigma), 292 mg Glutamine (Sigma), 10% FBS and 1% Penicillin Streptomycin (Life Technologies, Carlsbad, CA) were reconstituted in 500 ml dH$_2$O and supplemented with or without Asparagine at a concentration of 0 to 100 mg/L (A4159, Sigma).

### Customized shRNA proliferation screen

The screen was performed using the *Kras;p16p19*$^{null}$ RMS line T14R and the *Kras;p16p19*$^{null}$ NMS line Sca1-01. Cells were plated in DMEM supplemented with 10% FBS and 1% Penicillin-Streptomycin on day -1 and infected with lentiviruses in the presence of 8 μg/ml Polybrene (Millipore, Billerica, MA) on day 0. Infected cells were selected by adding Puromycin (Santa Cruz, Biotechnology, Dallas, TX) to a final concentration of 2 μg/ml on day +1. Cell growth was evaluated by CellTiter Glo (Promega, Fitchburg, WI) on day 8. RMS cells were plated at 900 cells per well and infected with 6 μl virus in 3 replicates exposed to Puromycin and 2 replicates maintained without Puromycin. NMS cells were plated at 450 cells per well and infected with 4 μl virus in 2 replicates exposed to Puromycin and 2 replicates maintained without Puromycin.

For each cell line, raw data obtained from cells exposed to Puromycin ( + Puromycin, y axis) were plotted against raw data from cells grown without Puromycin (- Puromycin, x-axis; *Figure 1B,G*). Raw data obtained from cells exposed to PGW or medium only were excluded from the analysis. Standard deviations (SD) from the mean were calculated for each data point, and those with SDs above the upper adjacent limit also were excluded from further analysis. Correlation coefficients between + Puromycin and - Puromycin data were 0.8348 (RMS) and 0.9501 (NMS), thereby confirming adequate transduction efficiency.

For each shRNA, replicate data were pooled and relative cell growth was quantified as the percentage proliferation of shRNA infected cells compared to the mean proliferation of cells infected with cntrl-shRNAs on the same plate. Receiver operator curve analysis using CTR01 data confirmed the ability of the screen to distinguish between the growth-inhibitory effects of shGFP and the neutral effects of cntrl-shRNAs (*Figure 1C,H*). For RMS and NMS, relative growth of less than 52% or less than 40% of cells infected with cntrl-shRNAs (light gray line in *Figure 1D,E*, and I, J), respectively, was associated with a false discovery rate less than 30%. Growth differences between cells subjected to silencing of one specific candidate gene and cntrl-shRNA-infected cells were tested for statistical significance using T-tests. Q-values were estimated using the algorithm published by J. W. McNicol and G. Hogan (*McNicol, 2013*). The growth-inhibitory effects of shRNA-mediated silencing of individual candidate genes were considered significant if $p<0.0001$ and $q<0.01$ and 3 shRNAs scored with an FDR<30%.

Candidate gene contributions to *Kras;p16p19$^{null}$* sarcoma growth were tested using a customized, in vitro shRNA proliferation screen designed using shRNAs from The RNAi Consortium (TRC) delivered in puromycin-selectable pLKO lentiviral vectors. The screen was performed in ten 96-well-plates (DAS36-DAS45, *Supplementary files 3–4*) using one shRNA per well. Five discrete shRNAs were used for each of the 141 candidate genes. The screen also included 3 shRNAs directed against KRAS (shKRAS; positive control) and control shRNAs directed against RFP, LUC or LACZ (cntrl; negative control). Additionally, two 96-well-plates (CTR01, *Supplementary files 5–6*) were infected with shRNAs directed against GFP (shGFP; silencing KRAS-IRES-GFP and thereby serving as a positive control) and control shRNAs directed against RFP, LUC and LACZ. Empty pLKO lentiviral vectors (designated PGW; did not contain shRNAs) and medium (medium; did not contain any virus) served as transduction and puromycin controls, respectively. TRC clone IDs and viral titers are listed in *Supplementary files 1–4.* The screen was performed using the *Kras;p16p19$^{null}$* RMS line T14R and the *Kras;p16p19$^{null}$* NMS line Sca1-01.

## Immunohistochemistry

Primary human sarcoma tissue was evaluated using commercially available sarcoma tissue arrays (SO2081, US Biomax, Rockville, MD). Human sarcoma sections were stained for ASNS (1 in 100, HPA029318, Sigma; human brain served as positive and muscle as negative control tissue). Mouse tumors were stained for ASNS (1 in 200, HPA029318, Sigma). Antigen retrieval was performed in 10 mM sodium citrate buffer pH6, and tissue sections were blocked in PBS, 5% BSA, pH7.4.

## RNA isolation and qRT-PCR

RNA was isolated from fetal and adult whole skeletal muscle, human RD, HT1080, Rh3, Rh5, Rh10, Rh28, Rh30, Rh41 and Rh36 cells by TRIzol extraction followed by DNAse digestion and purification using the RNeasy Plus Micro Kit. RNA was reverse transcribed using Superscript III First-Strand Synthesis System (Life Technologies) for RT-PCR (Invitrogen). qRT-PCR was performed using an ABI 7900 RT-PCR system (Applied Biosystem) with SYBR-green PCR reagents (Life Technologies). Human *ASNS* and mouse *Gls* detected using the following primer sequences (Eurofins MWG Operon, Huntsville, AL): GGAAGACAGCCCCGATTTACT (ASNS, fw), AGCACGAACTGTTGTAATGTCA (ASNS, rev), TTCGCCCTCGGAGATCCTAC (Gls, fw), CCAAGCTAGGTAACAGACCCT (Gls, rev).

## Western blot

Cells were lysed for 10 min on ice in 50 mM HEPES (4-(2- hydroxyethyl)-1-piperazineethanesulfonic acid), pH7.4, 40 mM NaCl, 2 mM EDTA, 10 mM sodium pyrophosphate, 10 mM sodium beta-glycerophosphate, 1% Triton X containing complete mini protease inhibitor cocktail (Roche, Indianapolis, IN), 50mM NaF and 1 mM sodium orthovanadate. Equal amounts of extract were processed for Western blot using rabbit polyclonal anti-ASNS antibody (1 in 250, HPA029318, Sigma). ASNS protein expression was evaluated using rabbit polyclonal anti-ASNS antibody (1 in 250, HPA029318, Sigma). Immune complexes were detected by chemiluminescence (ECL, 32132, Thermo Fisher Scientific, Waltham, MA).

## Proliferation assays

Sarcoma cells were exposed to asparaginase (0.1-10 IU/ml, stock 5 IU/ml in 0.9% NaCl, MyBioSource, San Diego, CA) and AS5 (50-500 mM, stock 10 mM in 0.9% NaCl, synthesized by Nigel G Richards). Proliferation assays were performed as previously described (*Hettmer et al., 2011*). Cells were plated at 1000–7000 cells per well on day -1 and exposed to chemicals or vehicle on days 0 and 2. Cell growth was determined by MTT assay (Cayman Chemicals, Ann Arbor, MI) on day 4 and quantified as fold-increase in MTT uptake compared with baseline. All assays were performed in triplicate and replicated in two to four independent experiments. Estimated EC50 concentrations were calculated using GraphPad Prism.

## ASNS silencing

ASNS expression was silenced using TRC shRNAs delivered in pLKO vectors. TRC clones TRCN0000324779, TRCN0000031703 and TRCN0000031702 were used to silence mouse *Asns*. TRC clones TRCN0000045875 and TRCN0000045877 were used to silence human *ASNS*. TRC clones TRCN0000033260 and TRCN0000033262 were used to silence *Kras* in mouse sarcoma lines. Control cells were infected with lentiviruses carrying TRCN0000072250 (shLUC) and TRCN0000072250 (shLUC) or TRCN0000072231 (shLACZ) and TRCN0000072240 (shLACZ) to control for off-target effects.

Mouse *Kras;p16p19$^{null}$* RMS and NMS cells were plated at 1000 cells per well on day -1, infected with lentiviruses in the presence of 8 µg/ml polybrene on day 0 and selected with puromycin at a final concentration of 2 µg/ml starting day 1. Effects of ASNS silencing were evaluated on days 3–5. Human Rh30 cells were plated at 5000 cells per well on day -1, infected with lentiviruses in the presence of 8µg/ml polybrene on day 0 and selected with puromycin at a final concentration of 0.5 µg/ml starting day 1. Transduced Rh30 cells were passaged and re-plated at 5000 cells per well. Effects of ASNS silencing were evaluated 3–5 days after replating.

## Annexin V staining

Annexin V staining was performed according to the manufacturer's instructions using Annexin V-APC (550474, BD Biosciences, Franklin Lakes, NJ) and PI.

## BrdU assay

Cells were incubated with 10 mM BrdU (552598, BD Biosciences, Franklin Lakes, NJ) for one hour at 37°C. The BD BrdU flow kit (552598, BD Biosciences) was used to fix and permeabilize cells prior to DNAse treatment and staining with anti-BrdU-APC (1 in 20, 17-5071-42, eBioscience, San Diego, CA) and Dapi (1 in 1000).

## OP-puromycin staining

OP-puromycin (MedChemExpress, Monmouth Junction, NJ) was reconstituted in PBS pH6.4 and stored at minus 20 degrees centigrade. Cells were incubated with 50 µM OP-puromycin for 1 hr at 37 degrees centigrade, fixed and stained with Alexa Fluor 555-Azide (Life Technologies) as previously described (*Liu et al., 2012*). Control cells were treated with 50 µg/ml cycloheximide for 15 min immediately prior to OP-puromycin exposure. Alexa Fluor 555 labeling was analyzed using an Olympus IX51 microscope at 20X.

In vivo asparaginase treatment. Asparaginase was reconstituted in 0.9% normal saline (NS) and stored at 4°C up to 3 weeks. Mice were treated daily with Asparaginase (Elspar, Lundbeck Inc, Deerfield, IL) by intraperitoneal injection at 1500 IU/kg (*Szymanska et al., 2012*).

## Serum amino acid levels

Amino acid levels in mouse serum were determined by HPLC.

## Statistics

Differences in cell growth, tumor growth, Annexin staining, BrdU staining and serum amino acid levels were tested for statistical significance using T-tests. Differences in tumor latency were evaluated by logrank (Mantel-Cox) test (ns $p \geq 0.05$, *$p < 0.05$, **$p < 0.01$, ***$p < 0.001$).

## Acknowledgements

We thank Francesca Izzo and Amy Schlauch in the DFCI RNAi Facility for technical assistance with the shRNA screen, Daniel Sherley and the DFCI pharmacy for providing pharmacy-grade Asparaginase (Elspar), Mark Kellogg at Boston Children's Hospital for mouse serum amino acid analysis and Joyce LaVecchio, Girijesh Burizula and Atsuya Wakayabashe in the Joslin Diabetes Center Flow Cytometry Core (supported by the Harvard Stem Cell Institute and NIH P30DK036836) for flow cytometry support. We are grateful to Tata Nageswara Rao, Richard Lock, Sean Morrison and Leonard Wexler for helpful discussions, and to Jun Hiratake and Hideyuki Ikeuchi for their contributions to the development of AS5. This work was funded in part by a Stand Up To Cancer-American Association for Cancer Research Innovative Research Grant (SU2C-AACR-IRG1111; to AJW), NIH grants P01 CA050661, P01 CA142536, U01 CA176058 (to WCH) and by a SARC-SPORE career development award and PALS Bermuda/St. Baldrick's (to SH). The authors declare no competing financial interests. Content is solely the responsibility of the authors and does not necessarily represent the official views of the NIH or other funding agencies.

## Additional information

### Competing interests

AJW: Reviewing editor, *eLife.* The other authors declare that no competing interests exist.

### Funding

| Funder | Grant reference number | Author |
| --- | --- | --- |
| American Association for Cancer Research | SU2C-AACR-IRG1111 | Amy J Wagers |
| Sarcoma Foundation of America | SARC-SPORE grant | Simone Hettmer |
| St. Baldrick's Foundation | Pediatric Research Grant | Simone Hettmer |
| National Institute for Health Research | P01 CA050661 | William C Hahn |
| National Institute for Health Research | P01 CA142536 | William C Hahn |
| National Institute for Health Research | U01 CA176058 | William C Hahn |

The funders had no role in study design, data collection and interpretation, or the decision to submit the work for publication.

### Author contributions

SH, Conception and design, Acquisition of data, Analysis and interpretation of data, Drafting or revising the article; ACS, WCH, Acquisition of data, Analysis and interpretation of data, Drafting or revising the article; DT, CLP, Acquisition of data, Drafting or revising the article; MS, Acquisition of data; RTB, NGJR, Analysis and interpretation of data, Drafting or revising the article; AJW, Conception and design, Analysis and interpretation of data, Drafting or revising the article

### Ethics

Animal experimentation: This study was performed in strict accordance with the recommendations in the Guide for the Care and Use of Laboratory Animals of the National Institutes of Health. All of the animals were handled according to approved institutional animal care and use committee (IACUC) protocols of the Joslin Diabetes Center. The protocol was approved by the Committee on the Ethics of Animal Experiments of the Joslin Diabetes Center. All surgery was performed under tribromoethanol or isoflurane anesthesia, and every effort was made to minimize suffering.

# Additional files

## Supplementary files

• Supplementary file 1. *Kras; p16p19*$^{null}$ RMS cell line T14R: raw data from shRNA screen (plates DAS36-DAS45).

• Supplementary file 2. *Kras; p16p19*$^{null}$ NMS cell line Sca1-01: raw data from shRNA screen (plates DAS36-DAS45).

• Supplementary file 3. *Kras; p16p19*$^{null}$ RMS cell line T14R: raw data from shRNA screen (control plate CTR01).

• Supplementary file 4. *Kras; p16p19*$^{null}$ NMS cell line Sca1-01: raw data from shRNA screen (control plate CTR01).

• Supplementary file 5. *Kras; p16p19*$^{null}$ RMS cell line T14R: statistical evaluation (Significance levels: 1, 5 shRNAs with FDR<30%; 2, 4 shRNAs with FDR<30%; 3, 3 shRNAs with FDR<30%).

• Supplementary file 6. *Kras; p16p19*$^{null}$ NMS cell line Sca1-01: statistical evaluation (Significance levels: 1, 5 shRNAs with FDR<30%; 2, 4 shRNAs with FDR<30%; 3, 3 shRNAs with FDR<30%).

• Supplementary file 7. Amino acid levels (µM) in Asparaginase-treated and untreated mice.

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
