## [Decision Letter]

Thank you for submitting your work entitled "Functional genomic screening reveals asparagine dependence as a metabolic vulnerability in sarcoma" for peer review at *eLife*. Your submission has been favorably evaluated by Fiona Watt (Senior Editor) and three reviewers, one of whom is a member of our Board of Reviewing Editors.

The following individuals responsible for the peer review of your submission have agreed to reveal their identity: Hideyuki Okano (Reviewing Editor) and Yoji Minamishima (peer reviewer).

The reviewers have discussed the reviews with one another and the Reviewing Editor has drafted this decision to help you prepare a revised submission.

The manuscript by Hettmer and colleagues presents the results of an shRNA-based proliferation screen in a *p1619^null^* KRAS (*G12v*) model of RMS and identifies asparagine synthethase (ASNS) as a main contributor to proliferation in the screened cells. It further shows a decrease in cell proliferation, increase in cell death and decrease of polypeptide synthesis after ASNS silencing in the model cells and human RMS cell line, effects that can be reversed by addition of exogenous asparagine.

All the experiments seemed to be well designed. All the data seems to be firm, have adequate controls and looks reliable. Although Asn requirement was already known in other malignant tumors such as ALL, it would be still valuable knowledge that Asn could be therapeutic target in sarcomas, which have fewer therapeutic options.

The authors need to address the following issues before the manuscript is acceptable for publication:

1) The authors clearly showed that Asn depletion suppressed nascent polypeptide synthesis, which might suppress tumor growth. If it is the cause of the tumor growth suppression, depletion of other amino acids would do the same. It would be very nice for the readers if the authors have some comments or discussion on why only Asn is required or why other amino acid metabolism pathway were not picked up by their screening. Furthermore, are the rescuing effects of Asn dose-dependent?

2) Similarly, ASNS reaction is coupled with Gln-Glu conversion. Asparaginase treatment or ASNS-knockdown might reduce Glu abundance. On the other hand, supplementation of Asn will interfere with ASNS enzyme activity, which might suppress Glu production. The authors should briefly comment on why Gln or Glu is not involved in this context. For example, Gln/Glu could be generated by other metabolic pathways, or something like that. Thus, evaluation of asparagine, aspartate, glutamine, glutamate levels before and after the treatment would be required.

3) Based on these data, the authors need to provide some mechanistic insight revealing why the combination of asparagine depletion and ASNS inhibition is required for inhibition of tumor growth in vivo. Is the inhibitory effect on polypeptide chain synthesis the only underlying mechanism for the observed metabolic vulnerability in sarcoma (i.e. tumor growth suppression)?

---

## [Author Response]

*1) The authors clearly showed that Asn depletion suppressed nascent polypeptide synthesis, which might suppress tumor growth. If it is the cause of the tumor growth suppression, depletion of other amino acids would do the same. It would be very nice for the readers if the authors have some comments or discussion on why only Asn is required or why other amino acid metabolism pathway were not picked up by their screening. Furthermore, are the rescuing effects of Asn dose-dependent?*

We agree with the reviewers that it is very well possible that sarcoma growth and survival require adequate availability of amino acids other than asparagine. Unfortunately, the design of the shRNA proliferation screen described in this study did not allow for comprehensive functional testing of genes encoding for enzymes relevant in amino acid biosynthesis. Instead, the screen was designed to evaluate the functional contributions of a particular group of sarcoma signature genes that were expressed at higher levels in both *Ras*-driven mouse sarcomas and human sarcomas. Within this cluster of sarcoma genes, top-scoring cellular functions were cell cycle control and cell division (Hettmer et al., 2011).

Nonetheless, in addition to ASNS, the candidate group of sarcoma genes we evaluated in this study did include 3 other transcripts encoding for enzymes relevant in amino acid biosynthesis. These enzymes were phosphoserine aminotransferase (PSAT1), phosphoglycerate dehydrogenase (PHGDH) and branched chain aminotransferase 1 (BCAT1). Recently published findings indicate that high expression of PHGDH in tumor tissue is a requirement for cell growth in epithelial malignancies (Locasale et al., 2011; Possemato et al., 2011), and both PHGDH and PSAT1 contribute to serine/glycine biosynthesis which plays an important role in supporting nucleotide synthesis in rapidly proliferating cells (Labuschagne et al., 2014; Jain et al., 2012). Branched chain amino acids provide anaplerotic substrates for the TCA cycle (Boroughs and DeBerardinis, 2015). However, in our screen, silencing of PSAT1, PHGDH or BCAT1 did not result in growth inhibition, suggesting that sarcoma cells show lesser dependence on these pathways.

To provide a more comprehensive discussion of amino acid metabolism pathways in tumor cell growth, we have revised manuscript to include the above considerations (Discussion, fourth and fifth paragraphs). We also included a new series of experiments documenting the effects of increasing concentrations of asparagine on the growth of shASNS and sh-LUC-transduced mouse *Kras;p16p19^null^* RMS and NMS cells. These experiments revealed that ≥ 10mg/L asparagine was needed to rescue the growth-inhibitory effects of ASNS silencing, whereas 0.01 – 1mg/L asparagine were insufficient. Interestingly, normal asparagine concentrations in mouse and human plasma are 3.8mg/L and 7.3mg/L, respectively (Cooney et al., 1970). Thus, even slight reductions in physiologic asparagine levels make sarcoma cells vulnerable to ASNS silencing (see Figure 2—figure supplement 2 and subsections “ASNS silencing inhibits growth of mouse *Kras;p16p19^null^* sarcoma cells by asparagine starvation” and”Asparagine depletion impedes mouse sarcoma growth in vivo”).

*2) Similarly, ASNS reaction is coupled with Gln-Glu conversion. Asparaginase treatment or ASNS-knockdown might reduce Glu abundance. On the other hand, supplementation of Asn will interfere with ASNS enzyme activity, which might suppress Glu production. The authors should briefly comment on why Gln or Glu is not involved in this context. For example, Gln/Glu could be generated by other metabolic pathways, or something like that. Thus, evaluation of asparagine, aspartate, glutamine, glutamate levels before and after the treatment would be required.*

Asparagine synthetase catalyzes the biosynthesis of asparagine from aspartate through an ATP-dependent transamination reaction. In this reaction, glutamine (Gln) acts as amino group donor and is converted into glutamate (Glu). As noted by the reviewers, ASNS inhibition or exogenous Asparaginase treatment should result in increased aspartate and glutamine, as well as decreased glutamate levels. As suggested, we confirmed this by measuring amino acid levels before and after treatment in Asparaginase treated mice ([Supplementary-material SD7-data]).

As the reviewer correctly points out, we cannot exclude that glutamate deprivation and/or aspartate excess negatively impact sarcoma cell survival/proliferation in addition to the growth inhibitory effects of asparagine withdrawal. However, increased glutaminase expression in cancer cells (Huang et al., 2014) could counterbalance glutamate reduction, especially in a glutamine-rich environment. Aspartate, on the other hand, carries reducing equivalents in the malate-aspartate shuttle (Chen et al., 2013), and increased aspartate levels after ASNS knockdown conceivably would benefit the redox state of sarcoma cells. Nonetheless, we recognize that disruption of Gln/Glu metabolism may represent an additional mechanism by which modulation of asparagine availability disrupts sarcoma cell growth.

We have addressed all of these important points in the revised manuscript, including an explicit discussion of the potential effects of altered glutamate/aspartate availability after Asparagine depletion (Discussion, third paragraph). We also demonstrated increased expression of Glutaminase transcripts in *Kras;p16p19^null^* sarcoma cells compared to normal mouse skeletal muscle (see Figure 2—figure supplement 3 and the aforementioned paragraph), which, as noted above, may counterbalance glutamate reduction due to ASNS inhibition.

*3) Based on these data, the authors need to provide some mechanistic insight revealing why the combination of asparagine depletion and ASNS inhibition is required for inhibition of tumor growth in vivo. Is the inhibitory effect on polypeptide chain synthesis the only underlying mechanism for the observed metabolic vulnerability in sarcoma (i.e. tumor growth suppression)?*

In our experiments, ASNS silencing in sarcoma cells did not impede sarcoma growth in vivo unless systemic asparagine was depleted by treatment with asparaginase. One could argue that limited nutrient availability within a rapidly enlarging solid tumor and slow asparagine uptake speak against repletion of tumor cell asparagine content from systemic sources. Nevertheless, our data support the notion that exogenous asparagine in the cell environment restores intracellular asparagine availability after ASNS silencing. This is discussed more extensively in the revised manuscript (Discussion, first paragraph).

Up until recently, the only known use of asparagine in mammalian cells was in protein synthesis, Asparagine does not contain the reducing equivalents of certain other amino acids, and Aspartate consumption due to Asparagine biosynthesis may diminish the reducing power of the malate-aspartate shuttle (Boroughs and DeBerardinis, 2015. Metabolic pathways promoting cancer cell survival and growth. Nat Cell Biol, 17, 351-9.; Chen et al., 2013; Huang et al., 2014). However, we agree with the reviewers that impaired peptide biosynthesis may not be the only mechanism responsible for the asparagine dependence of sarcoma cells observed in this study. Recently published data revealed that, surprisingly, asparagine supplementation suppressed cell death in glutamine-deprived cells (although it did not restore TCA anaplerosis and proliferation) (Huang et al., 2014). Thus, Asparagine appears to promote cellular adaptation to metabolic stress such as glutamine depletion. Zhang et al. also noted that asparagine is the last amino acid synthesized in the TCA cycle and that its amination exclusively depends on glutamine (Huang et al., 2014). Amino acid availability is known to stimulate mechanistic target of rapamycin (mTOR) complex 1, which integrates environmental and intracellular signals to regulate cell growth (Jewell et al., 2015). Taken together, these observations by ourselves and others suggest that Asparagine may serve a central role as a cellular sensor of TCA cycle intermediate/reduced nitrogen availability and, ultimately, as a metabolic regulator of cell behavior.

An in-depth discussion of potential other mechanisms responsible for sarcoma cell asparagine dependence was added to the revised manuscript (Discussion, fourth paragraph).